# "Are all referrals necessary?" Experiences and perceptions of maternity healthcare providers on emergency intrapartum referrals in Dar es Salaam, Tanzania

**Shekha Selemani**[ID][1]*, **Michael O. Mwakyusa**[1], **Selemani Bashiri**[1], **Mangi J. Ezekiel**[2], **Dorkasi L. Mwakawanga**[ID][3], **Fadhlun M. Alwy Al-beity**[1], **Andrea B. Pembe**[1]

1 Department of Obstetrics and Gynecology, Muhimbili University of Health and Allied Sciences, Dar es Salaam, Tanzania, 2 Department of Behavioral Sciences, Muhimbili University of Health and Allied Sciences, Dar es Salaam, Tanzania, 3 Department of Community Health Nursing, Muhimbili University of Health and Allied Sciences, Dar es Salaam, Tanzania

* shekha_alinisi@yahoo.com

## Abstract

### Background

Intrapartum continuity of care to reduce maternal morbidity and mortality relies heavily on a functional and effective referral system between tiers of care. Capacity building of providers in managing intrapartum referrals is expected to improve the efficiency of the referral system, but this does not always work in practice. This study explored the experiences and perceptions of maternity healthcare providers on emergency intrapartum referrals in Dar es Salaam, Tanzania.

### Methods

An exploratory qualitative study was conducted at Amana Regional Referral Hospital and Muhimbili National Hospital in Dar es Salaam. Maternity healthcare providers were purposively recruited based on cadre, working experience of more than three years in the maternity wards. An in-depth interview guide which involved questions and probes was used to conduct eleven interviews. Data was thematically analyzed.

### Results

Three major themes emerged, namely: 1) causes of referrals are beyond medical indications; 2) limited maternity healthcare provider capability at referring facilities; and 3) limited communication between referring and receiving facilities. According to maternity healthcare professionals, referrals were seen as a way to minimize blame and a clinical management tool to prevent difficulties. They advocated for more understanding of the skill set among maternity healthcare providers, but some had negative perceptions towards performing their responsibilities.

**Data Availability Statement:** Data can not be shared publicly because they contain sensitive participants' informations and participants did not

consent to having their interviews shared. Data are available from the Directorate of Research and Publications at the Muhimbili university of health and allied sciences (contact via drp@muhas.ac.tz) for researchers who meet the criteria for access to confidential data.

**Funding:** The author(s) received no specific funding for this work.

**Competing interests:** The authors have declared that no competing interests exist

## Conclusions

Skills gaps among maternity healthcare providers at referring hospitals influenced referral decision-making and service provision. There was hostility between referring and receiving hospitals. Capacity-strengthening strategies such as ongoing skills training and changes in attitudes toward referrals require improvements. The referring hospital should only consider referrals as a last resort after other case management has been completed.

## Introduction

The maternal mortality rate in Tanzania has reduced from 760 to 238 per 100,000 live births in the last 2 decade [1]. This is a significant improvement after two decades of stagnant rates. Most maternal deaths occur from complications during labor, delivery and the immediate postpartum period [2]. Most women in Tanzania deliver in lower-level (primary health care) facilities with limited resources to care for complications during labor and delivery, sometimes necessitating emergency referrals to higher-level (secondary or tertiary care) centers [3]. The timeliness and efficiency of these emergency intrapartum referrals are key to saving women and their newborns from severe morbidities and death.

The World Health Organization's (WHO) Quality of Care for maternal and newborn health framework (Fig 1) includes a domain on the functionality of referral systems [4, 5]. There are several quality indicators to ensure timely and efficient referrals of women and newborns. These include prompt assessments and decisions to refer are made in a timely manner; availability of an established referral plan; and appropriate information exchange and feedback between the facilities involved (Fig 1) [6, 7]. In addition, the referred woman should be accompanied by competent personnel who will continue with the right level of care during transit and until arrival at the higher-level receiving facility.

There are many reasons for intrapartum referrals, including the complexity of medical conditions as well as gaps in skills, medical supplies and equipment available at a primary health care facility to manage complications. Other reasons include health workers' attitudes and confidence in tackling the complications, especially when they do not have adequate technical and infrastructural support Barriers to a functional referral system are contributed by a dysfunctional health system, such as lack of functional ambulances, inadequate number of skilled providers to cater for a large number of patients, no skilled personnel to accompany a referral, ambulances being unavailable or without adequate fuel [8].

Referrals are often seen as the 'safest' option for healthcare professionals to avoid maternal and fetal morbidity and mortality occurring at their own facility [9]. A study conducted in rural Zambia showed minimal consensus on decision-making for some obstetric conditions prior referral [10]. Moreover, high demand for delivery care services at primary health care facilities put health providers under pressure, which may influence the number of referrals to tertiary hospitals [11]. A qualitative study by Afari et al. in Ghana among maternity care providers found that perceived gaps in their skill sets led them to refer patients to tertiary hospitals [12]. Studies have been done in Tanzania on the perceptions of women and maternal care providers on the causes, effectiveness and quality of the maternal referral system and obstetric care in referral hospitals. However, all these studies were conducted in rural settings in Tanzania [13–15]. Little is known about the perceptions and experiences of maternity care providers on emergency intrapartum referrals in urban settings. Therefore, this study reports the perceptions and experiences of maternity care providers on emergency intrapartum referrals at a regional referral hospital and a tertiary hospital in Dar es Salaam City, Tanzania.

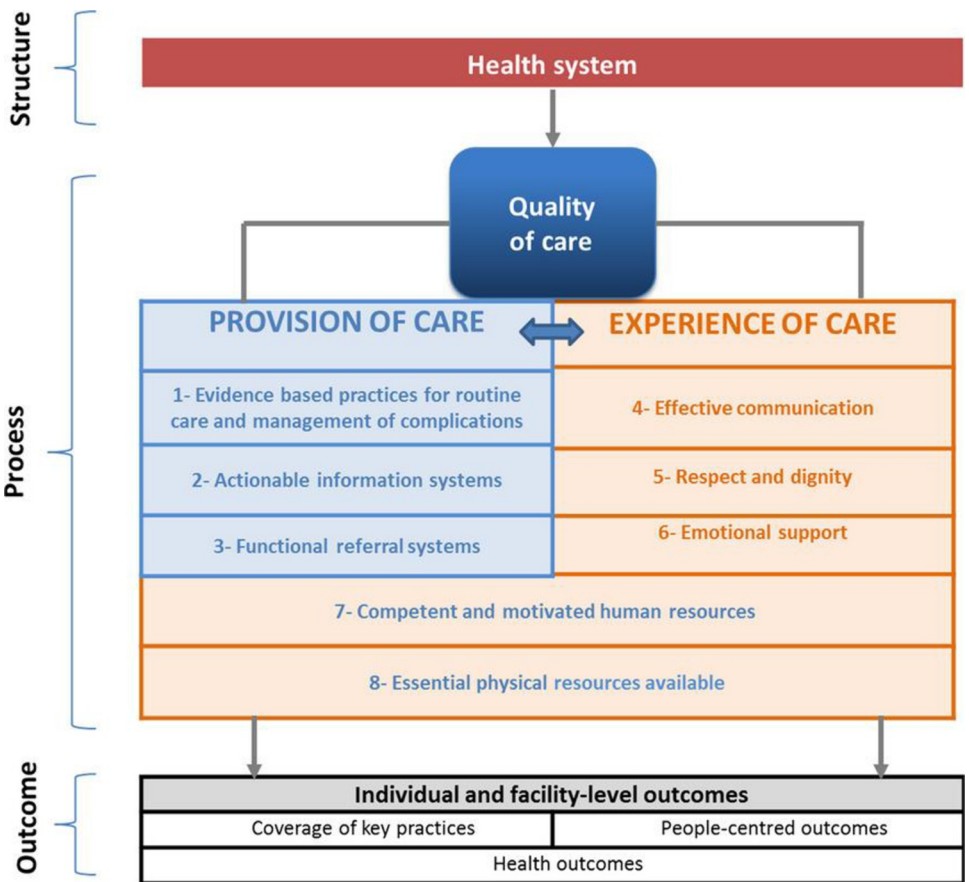

**Fig 1. WHO quality of care framework for maternal and newborn health.**

The illustration (Fig 2) shows constructs that may influence the perception and experience of maternity care providers on emergency intrapartum referrals. Since we could not find an existing framework for examining the perceptions and experiences of maternity care professionals about emergency intrapartum referrals, we adapted a framework from a relevant study to develop a framework for this study [16]. The framework of this study (Fig 2) demonstrates that availability and usage of Standard Operating Procedures; patients' demands and satisfaction; patients' attitudes towards maternity care providers; perception of referrals as the safest option for healthcare providers; gaps in skill sets; minimal training; and the working environment (such as long working hours, heavy workload, weak supervision, poor communication with co-workers, and insufficient salaries) are among the factors that may influence perceptions of maternity healthcare providers towards emergency intrapartum referrals.

## Materials and methods

### Study design and setting

This explorative qualitative study used in-depth interviews to explore maternity healthcare providers' perceptions and experiences on emergency intrapartum referrals in the Dar es Salaam region. The study was done in Dar es Salaam, Tanzania's largest city, with a population density of about 5.4 million [17]. There are specific contextual challenges associated with an urban context in addition to challenges that exist as part of the health system.

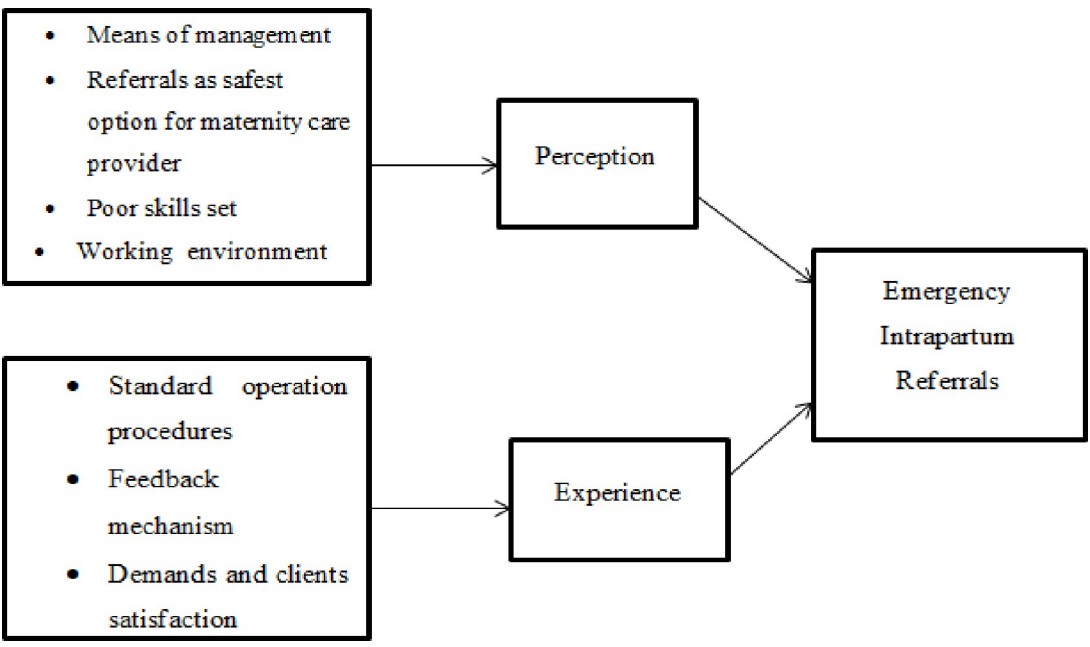

**Fig 2. A conceptual framework.**

The Tanzania health system is organized in a pyramidal structure where fewer higher-level hospitals support a network of lower-level facilities [18]. District hospitals serve as the first referral level for their immediate lower-level facilities, health centers and dispensaries. At this level, patients needing advanced services are transferred to the secondary level hospitals at the regional level and then, if needed, on to the tertiary level, which includes zonal, specialized, and National hospitals.

Some national guidelines and protocols describe the referral process; however, these are not widely known or followed. This study was done in two referral levels, Amana Regional Referral Hospital (ARRH), a secondary-level facility, and Muhimbili National Hospital (MNH), a tertiary-level facility. The two hospitals are both located in urban areas and have some similarities and some differences in characteristics in capacity, human resources, and infrastructures, as seen in **Table 1**.

## The referral process from Amana Reginal Referral Hospital to Muhimbili National Hospital

At ARRH: Once a referral decision has been made, the specialist on duty or the head of the department is consulted to examine the patient and authorize the referral. The nurse supervisor contacts Muhimbili National Hospital, the receiving facility, via telephone. They describe the patient's diagnosis, condition, reason for referral, and management already given. The supervisor organizes the ambulance and a nurse escorts the woman to the higher-level facility. Private cars are used to transport patients when the ambulance is unavailable due to lack of fuel or absence of a driver. There is a cost associated with renting a private ambulance, usually covered by the woman's family.

At MNH: the mobile phone for referral management stays with the specialist obstetrician on duty, who communicates with the referring doctor at ARRH. The obstetrician enquires about the case and the critical status of the patient and advises on any management to be given

**Table 1. Similarities and differences between the two hospitals in the study.**

| Characteristics | Amana Regional Referral Hospital [ARRH] | Muhimbili National Hospital [MNH] |
|---|---|---|
| Referral level | Secondary level, receiving referrals from district hospitals and large health centers | Tertiary level, receiving referrals from three secondary level hospitals in Dar es Salaam (Amana, Mwananyamala and Temeke), as well as from nearby Pwani region's hospitals such as Bagamoyo, Mkuranga and Kisarawe. |
| Infrastructure | Separate maternity building with antenatal, postnatal, neonatal, and labor wards with six delivery beds | Separate maternity building with antenatal, postnatal, neonatal, and labor wards with six delivery beds |
| | No separate Intensive Care Unit or High Dependency Unit | A dedicated Intensive Care Unit and High Dependency Unit within the maternity building |
| | Dedicated obstetric theatre with two operating rooms within the maternity building | Dedicated obstetric theatre with four operating rooms adjacent to the maternity building |
| Human resource | Seven medical specialists (obstetric/gynecologist), six registrars, and no residents | Twenty-six medical specialists (obstetrics/gynecologists, and consultants), sixteen registrars, and more than thirty residents who are medical doctors specializing in obstetrics/gynecology. |
| | 65 midwives of different levels of training (certificate to master's level) | 159 midwives of different levels of training (diploma to master's level) |
| | 13 Anesthetists | 8 Anesthetists per shift in obstetric theatre |
| Daily delivery load | 39–45 total deliveries per day | 25–35 total deliveries per day |
| Patient flow | A patient can be admitted directly from home but hospital also receives referrals from lower-level facilities within the district. | Patients can be admitted directly from home but hospital also receives referrals from primary- and secondary-level facilities |
| | | Muhimbili National Hospital also has an Intermural Private Practice. |

pre- and during referral, to stabilize the patient until they reach MNH. The obstetrician can also accept or redirect the referral to MNH-Mloganzila, which is another branch of the MNH.

When a woman is referred with an intrapartum emergency, she is received at the MNH labor ward, where initial management, including resuscitation, if necessary, is done. The on-call medical team reassesses and decides on the care plan, including whether she needs further consultations with senior doctors, interdepartmental consultations, or surgical management.

## Study participants and recruitment

A purposive sampling technique was used to attain maximum sample variability. We recruited maternity healthcare providers of different cadres and training levels, as well as maternity ward administrators. The administrators were the nurse-midwife and doctor in charge of the maternity ward, as well as the head of the department, from each of the two facilities. These groups of participants were selected as they are involved in managing emergency obstetric cases and implementing emergency intrapartum referral policies. We requested the maternity in charge in the two health facilities provide a list of all maternity care providers, to create a sampling frame of providers. The purposive selection of providers was based on the following criteria: three years or more experience in the department (to ensure experience on the referral system), cadre, and their roles in the referral process. Eleven participants including four doctors, five midwives, and two administrators (in charge of maternity wards, head of the department) working at Amana regional referral hospital and MNH participated in the study. The data collectors contacted the participants identified as eligible for interview either in person or by phone to establish rapport and set a day and time for an appointment to participate in interviews.

## Data collection

The data were collected in the two hospitals using an in-depth semi-structured interview guide prepared in English and then translated into Kiswahili. The interview guide contained a set of

main questions and then probing questions to give space for the participant to provide more responses or clarification to what has been said. The first author conducted interviews with the assistance of two research assistants who took notes and audio-recorded the interviews. Research assistants had training in sociology and were well-versed in qualitative data collection. Written informed consent was obtained from each participant for taking part in the interview and for the audio recording. Interview procedures were explained, and each participant was informed that his/her participation was voluntary and that they may choose not to answer some of the questions or stop the interview at any time. The in-depth interviews took place in a quiet private room within the participants' respective facilities, at a convenient time suggested by them. All interviews were conducted in Kiswahili, the native language spoken comfortably by data collectors and study participants. All interviews were audio-recorded and lasted between 45 and 60 minutes. The interviews stopped after data saturation was observed when no new information was obtained from the study participants.

## Data processing and analysis

The analysis was done concurrently with data collection. The analysis began after the completion of the first two individual interviews. Audio recordings of the interviews were transcribed verbatim and then translated into English. Thematic analysis was done as described by Braun and Clarke [19].

The translated transcripts were read through several times to familiarize with the data and to get an understanding of the participants' accounts. The first and third authors developed the initial coding scheme based on the study objectives, interview guide, and conceptual understanding of the referral system in Tanzania. To ensure reliability, they manually coded the first transcript independently and then compared the codes for agreement on the final codes and coding. A list of potential and initial codes reflecting the predetermined themes was created through data reduction. Themes and sub-themes were then generated from the general list of codes created through comparisons. Lastly, themes were reviewed and discussed by all authors, discrepancies were noted, and identified themes and sub-themes were finalized.

## Ethical approval and consent to participate in the study

Ethical clearance was obtained from the Muhimbili University of Health and Allied Sciences Institutional Review Board (MUHAS-REC-10-2021-867). The permission to conduct the study was obtained from the Director of Clinical Services at Amana Regional Referral Hospital and the Director of Clinical Services at Muhimbili National Hospital. Study participants were informed about study goals and objectives and were asked for permission to digitally audio-record interviews. Information provided by the participants was kept confidential and used solely for the research purpose. All the voice recordings of interviews were deleted after transcription and translation of the information. It was guaranteed to participants as part of the informed consent process that information was treated confidentially, anonymized before publication, and that quotes could not be traced back to individuals. We obtained written consent from all participants in the study.

**Methodological considerations.** Trustworthiness in a qualitative study is attained when the findings of such a study are worth believing [20]. Four criteria as explained by Shenton et al. were used to assess the trustworthiness in our study; credibility, dependability, transferability, and conformability [21]. The credibility of findings was ensured through engagement and cross-checking of themes, sub-themes, and findings among research team members. In addition, during the data collection phase, we spent a considerable amount of time interacting with study participants in both hospitals, in order to get a better understanding of referral

dynamics. The sharing of findings within the research team and discussion to agreement where there were differences of opinions was done to ensure that our results are dependable. Our results are transferable to other contexts since the findings section provides descriptions of the procedures, settings, and methods and therefore lend themselves relevant to other/similar contexts. Our study presents perspectives and experiences on referral from different cadres of staff din the two hospitals and therefore represents a fair range of differing viewpoints (authenticity) on issues around obstetric referrals in Tanzania.

The study achieved rich data through the involvement of maternity health care providers with experience in the provision of care and referrals in the facilities under study. Furthermore, it provided an in-depth perspective of providers concerning intrapartum referrals thereby identifying gaps and areas for improvement. Despite all these strengths, the study had some limitations. A major limitation is the fact that an assessment of referral guidelines was not done, so adequacy of the referral process to agreed standard could not be done per standard. Therefore, our results should be interpreted in light of this limitation.

## Results

We interviewed 11 participants with an age range of 29–56 years. The average age of the participants was 40 years. Just under half (45%) were female and just over half (55%) were male. Among them, five were midwives, two were registrars, three were obstetricians and gynecologists, and one was an administrator. All of them had working experience of at least three years.

Three main themes emerged from the analysis: (1) Reasons for emergency intrapartum referral are beyond medical indications (2) Limited maternity healthcare provider capability at referring facility (3) Limited communication between referring and receiving facilities. Each main theme included 2–5 sub-themes. See **Table 2** for an outline of themes and sub-themes.

### Theme 1. Reasons for emergency intrapartum referral are beyond medical indications

This theme explores experiences of participants on how the decision for referral is made and reasons for referrals. Five sub-themes are described under this theme: referral process, referrals due to a mismatch between delivery load and available facilities, referrals due to infrastructure challenges, referral to avoid blames, and lack of motivation.

**The process of referral.** The majority of referrals made were emergency intrapartum referrals. Several health workers reported not following the standard operating procedures during referrals. Women were being referred without getting the required pre-referral management. Commonly women or the accompanying people were not given the appropriate

**Table 2. Themes and sub-themes.**

| Themes | Sub-Themes |
|---|---|
| Reasons for emergency intrapartum referral are beyond medical indications | • The referral process<br>• Referrals due to a mismatch between delivery load and available facilities<br>• Referrals due to challenges in infrastructure<br>• Referrals to avoid blame<br>• Lack of motivation |
| Limited maternity healthcare provider capability at referring facility | • Lack of basic knowledge and skills in practices among maternity healthcare providers (MHP)<br>• The irresponsibility of MHPs at referring facility |
| Limited communication between referring and receiving facilities | • Harsh treatment to an escort nurse<br>• Lack of constructive feedback to the referring facilities |

documentation or a completed referral form. Several participants reported that an increase in the number of night referrals compared to referrals received during daytime drove this issue, partly due to a lack of supervision of junior staff during the night shifts. As one participant from receiving hospital shared:

"... A patient was brought with hemorrhage; she has 18–20 size cannula and was resuscitated with 5 liters of fluids. No one remembered to insert a urine catheter. And the patient continues to bleed... one cannot comprehend this...is this laziness or ignorance of the accompanying nurse..." (Administrator, receiving hospital)

A response from a participant at referring hospital was:

"... sometimes we receive many patients at once who need to be referred. Following standard operating procedures before referral might be difficult. Strictly speaking any referral must be issued by the doctor upon agreement, and a referral form will be written and signed" (Overall in-charge, referring hospital).

**Referrals due to a mismatch between delivery load and available facilities.** It was believed by participants that the overwhelming number of patients at referring facilities influenced the decision to make referrals. The hospital's infrastructure, including its operating rooms, neonatal and maternal intensive care units, and blood and transfusion product availability, precludes it from functioning as a referral hospital. This discourages it from providing referrals as a means of receiving assistance. A participant reported:

"... Right now, we have a big number of patients that we receive as emergencies but the environment itself in this hospital is not suitable to receive all of them..." (Specialist, referring hospital)

Another participant was quoted as saying:

"... A patient referred from [XX] hospital to [YY] hospital and straight was referred to Muhimbili hospital, which shows [YY], was overwhelmed and could not take any additional referrals..." (Midwife, receiving hospital).

**Referrals due to challenges in infrastructure.** There was a challenge with availability of infrastructure such as intensive care units, high dependency units, enough operating rooms, blood supply for transfusion, and an ambulance. When more than one emergency arose, this prompted the need for a referral to be made. In addition, because of the availability of one ambulance, then they preferred to call for a referral to Upanga (a nearby national hospital branch) compared to Mloganzila (a distant national hospital branch). To drive this point home, a participant was quoted as saying:

"When I was on duty yesterday, there was an urgent need for treatment for both an ectopic [pregnancy] patient and a PPH [post-partum hemorrhage] patient, so I was compelled to take the PPH patient into the operating room while sending the ectopic patient to a different hospital because there was no space to serve both patients" (Registrar, referring hospital).

Even when a correct decision to refer is made early, the actual referrals may take a long time (sometimes more than six hours). Participants narrated that facilities have one ambulance

and it takes a long time to reach a patient. More time is also lost when mobilizing funds for fuel or deciding the route. As one participant from a receiving hospital narrated bitterly:

*"A referral was made early for a woman with fetal distress or two prior scars in labor. She arrives at our hospital 6 or 8 hours later. They had difficulties in finding an ambulance or an escort nurse who runs between hospitals." (Midwife, receiving hospital)*

**Sometimes health workers refer patients to avoid blame.**    When a patient or patient's relative is a member of the medical profession or has government influence, they frequently demand a referral for no valid reason. The majority of these obstacles are faced by hospital administrators who must endure the burden of receiving phone calls from patients or their relatives demanding a referral:

". . .But when a relative wants or is in favour of getting a referral to her relative and then you decide to retain a patient, when things go wrong, you can take the blame" (Administrator, referring hospital).

Another participant echoed this point, saying:

". . .we sometimes give a referral to the patient whom we could manage here, only to avoid delays and blames. . ." (Registrar, referring hospital).

**Lack of motivation.**    It was reported when providers did not receive their allowance payments on time, their motivation decreased, resulting in sub-optimal performance. Some doctors had a decreased desire to manage patients who visited a referring hospital during their shift. In order to rest, he/she would choose to see some of the patients and refer other patients. Participants reported that other doctors routinely disregard standard operating procedures and refer patients without a genuine reason. A participant mentioned:

*"We receive some referrals without an informing call and no convincing reason of referral, some doctors will just refer the patient knowing when the patient reached receiving hospital will never be rejected and taken back to our hospital" (Registrar, receiving hospital).*

## Theme 2. Limited maternity healthcare provider capability at referring facility

This theme illustrates the perception of MHPs' capability at the referring hospital. Two subthemes were identified: lack of basic knowledge and skills MHPs and irresponsibility in treating patients.

**Lack of basic knowledge and skills in practices among MHPs at referring facility.**    Inadequate knowledge and skills among maternal care providers were found to exist, and some were incompetent in performing their responsibilities. Participants reported that when junior doctors failed to consult their seniors, they ended up making wrong decisions that consequently caused delays in the provision of management to the patient:

*". . . Most of the time there is a gap, you find an intern doctor can't decide and still does not consult. At the end, decisions for referral are made very late" (Specialist, referring hospital)*

A participant from a receiving hospital was quoted as saying:

*"...as I was working in the labor ward, there was a patient referred with a diagnosis of a previous scar with impending rupture, but after our investigation she was found to have umbilical hernia..." (Midwife, receiving hospital)*

**The irresponsibility of MHPs at the referring facility.**   Participants at the receiving hospital perceived the MHPs at the referring hospital as irresponsible since they sometimes referred patients unnecessarily. They further expressed that, sometimes during phone calls, they mention planning to refer two or three patients but then in practice they end up bringing four to five patients, just because they want to lessen their own workload. One participant said:

*"... maybe they receive many patients, and they fail to attend all of them. You find there is no genuine reason to refer them. They call for referral of two patients and they bring 4 of them..." (Overall in-charge, receiving hospital).*

## Theme 3. Limited communication between referring and receiving facilities

This theme illustrates the communication challenges among MHPs from the referring and receiving hospitals. These include harsh treatment of the MHPs by the referring hospital and lacking constructive feedback from the receiving hospital to the referring hospital.

**Harsh treatment to an escort nurse.**   Participants state that an escort nurse who arrived with the patient at receiving hospital sometimes was less informed about the condition of the patient and the referral form was not filled properly. The predicament led the maternity staff at the receiving hospital to act hostilely against the escort nurse. A participant reported:

*"... I was called to escort the patient while I was in another ward; it was hard to know the entire patient's information, especially if there were many patients. When you reached there, the staff acted haughtily and asked questions as if we didn't want to fulfill our obligations, even though that is not the case..." (Midwife, referring hospital).*

**Lack of constructive feedback to the referring facilities.**   Participants reported that feedback from the receiving hospital was only occasionally given. Sometimes it was given as a report which tends to end up at the administration level and not reach the maternity healthcare providers. For mismanaged cases which ended in maternal death, feedback was given through phone calls:

*"I was thinking there should be a procedure of sending feedback to the maternal care providers on the patients they referred and their outcome, we do write weekly reports, but it ended up at the managerial level and not reaching to the providers" (Midwife, receiving hospital)*

When a woman with an intrapartum emergency was referred to a tertiary hospital, she was received at the labor ward, where initial management, including resuscitation, was initiated. The on-call medical team reassesses and would decide on modality of care: whether she needs further consultations with senior doctors as inter-departmental consultations or surgical management within the maternity department. Unfortunately, no feedback information was sent back to the referring hospital regarding the progress of the management of the patient. A participant from referring hospital had this to say:

*"...They hardly ever bring us feedback or let us, the providers, receive it. Feedback rarely reaches the service providers ..." (Registrar, referring hospital)*

## Discussion

This study explored the perceptions and experiences of maternity care providers regarding intrapartum emergency referrals in a regional referral hospital and Muhimbili National Hospital in Dar es Salaam, Tanzania. Through in-depth interviews, we found out that referrals for reasons beyond medical indications (such as referring a patient to avoid blame and deaths occurring at a particular health facility). There were also gaps in competencies and irresponsibility of maternity healthcare providers; as well as limited communication between a referring and receiving health facility (such as lacking constructive feedback and use of harsh language to escort health personnel) were to be the main findings.

Our study found that decision-making for the provision of referrals was influenced by high patient demand for quality service delivery. When a patient was health personnel or had influence, there was pressure on maternity health care providers to provide a referral without an apparent reason. Since everyone wishes to receive the best care and most patients believe the national hospital provides the best care, they demand referrals even without a genuine reason. Studies conducted in Indonesia and Tanzania [9, 11] found that providers perceived patients to be overly demanding during the. In addition, the patient's poor perceptions of the hospital's services places them (patients) under duress and also has an effect on the referrals made. The relationship between the patient and provider is crucial to the delivery of quality services, and requests for referrals may strain that relationship.

The intrapartum period is a very crucial period that requires close care and monitoring to ensure the safety of both the mother and the baby. A similar finding with the study, showed meeting patient demands and prevent adverse outcomes, maternal care providers often refer patients as a way to avoid blame, and sometimes make referrals without communication with the receiving facility beforehand [11]. A lack of pre-referral communication, the high number of patients at referring hospitals, and the absence of feedback all contribute to an increase in intrapartum referrals and a dysfunctional referral system [22–24].

Similar findings on the referral to avoid delays and blames were found in a Sierra Leone study, [12] which showed that some referrals were made to avoid complications or maternal death at their facility. Maternal care providers may refer a patient to avoid liability in the event of adverse outcomes or after receiving negative feedback. This could be explained by fear of damaging their personal or institutional reputation if a negative outcome happened.

The unfriendly working environment at referring hospital lacking intensive care units and high dependency units for adults and neonates, a sufficient blood supply for transfusion, a sufficient number of operating rooms, and sufficient staff) plus gaps in timely payment of allowances, lowered many MHPs' motivation at work. Similar studies report staff shortage as a significant challenge influencing referrals [11, 15, 25]. Studies done in Ghana, Ethiopia and Indonesia [11, 15, 26] revealed similar findings regarding the need for more sufficient infrastructure and staff to avoid unnecessary patient referral. The unconducive working environment contributed significantly to stress, frustrations, fatigue, low job satisfaction, and loss of morale. When providers are not paid their allowances on time, their morale to work at a high level is diminished. This could be because the government has upgraded these hospitals to referral hospitals, where the numbers of patients and workload have increased, but the working environment and staffing levels remained the same. It brought the challenge of increasing the number of referrals to the tertiary hospitals. This reduced the quality of services provided at tertiary hospitals because of needing more support.

In our study, some providers needed to gain skills in managing patients. These providers, who may be newly employed, interns, or those who were resistant to learning, caused delays in providing proper management or made incorrect referral decisions. This could be explained

by the need for adequate supervision from their seniors, who were found to be typically not available at night. Similar studies reported that inadequate training, which leads to a lack of skills and inadequate compensation, facilitates maternal health care providers to refer patients to other facilities [16]. Other studies [12, 27] showed that gaps in clinical competency influenced providers in referring women with intrapartum emergencies. Some colleagues were incompetent in making patients' diagnosis because he/she had little work experience compared to senior providers; and consequently, tended to refer more patients due to a lack of skills and confusion regarding clinical referral criteria [8, 26]. The similarity of these observations may be due to the lack of motivation of the seniors for not being paid their allowances in a timely way, so they opt to work during the daytime and use the other time for part time jobs to supplement their income.

Feedback to referring facilities provides an opportunity for providers to learn where they did wrong or the proper way to handle various cases, thereby improving the quality of services. According to our study, feedback was only sometimes provided, and there was no formal way of providing feedback. This could be a weakness on the part of the administration, wherein responsible personnel failed to perform their duties or raise a challenge (if they had any) regarding the delivery of feedback. Other studies have highlighted the lack of regular feedback to referring facilities and inconsistent use of feedback forms, instead, only complaints about what needed to be appropriately done or expected to be done were reported [22, 28]. Other studies with comparable findings on referral feedback approaches demonstrated that referring facilities were not always provided with feedback, and that feedback was sometimes given to the women or other close family members instead to maternity healthcare providers [29]. The lack of standard national referral guidelines had made no uniformity in delivering services, especially in giving feedback which is a barrier to a functional referral system [24].

## Conclusion

Our study found that there was referral hostility between referring and receiving hospitals. The skills gap among maternal care providers at referring hospitals still existed which influencing the decision-making and provision of quality services. Referral should be regarded as the last option after the patient has been managed to the greatest extent possible at the referring hospital, and pre-referral management should be provided before referral. Strategies to strengthen communication between the hospitals are warranted and ongoing training should be emphasized. Standard referral guidelines should be developed so that each hospital knows their roles and expectations. Further studies on the experiences of women and their relatives on referral services should be conducted to get their insights into where the systems are lacking and what needs to be done.

## Acknowledgments

The authors acknowledge the support received from all the participants in this study. We also thank Amana Regional Referral Hospital and Muhimbili National Hospital for granting access to use their facilities in this study.

## Author Contributions

**Conceptualization:** Shekha Selemani, Mangi J. Ezekiel, Andrea B. Pembe.

**Data curation:** Shekha Selemani, Michael O. Mwakyusa, Selemani Bashiri.

**Formal analysis:** Shekha Selemani, Dorkasi L. Mwakawanga, Fadhlun M. Alwy Al-beity.

**Methodology:** Shekha Selemani, Mangi J. Ezekiel, Andrea B. Pembe.

**Supervision:** Mangi J. Ezekiel, Andrea B. Pembe.

**Writing – original draft:** Shekha Selemani.

**Writing – review & editing:** Shekha Selemani, Michael O. Mwakyusa, Selemani Bashiri, Mangi J. Ezekiel, Dorkasi L. Mwakawanga, Fadhlun M. Alwy Al-beity, Andrea B. Pembe.

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
