## [Decision Letter · Decision Letter 0]

13 Mar 2023

PONE-D-22-34478“Are all referrals necessary ?” Experiences and perceptions of maternity healthcare providers on emergency intrapartum referrals in Dar es Salaam, Tanzania.PLOS ONE

Dear Dr. SHEKHA SELEMANI,

Your manuscript entitled “Are all referrals necessary ? Experiences and perceptions of maternity healthcare providers on emergency intrapartum referrals in Dar es Salaam, Tanzania" which you submitted to Plos One Global Health Journal, has been reviewed. After careful look at the paper, we are willing to consider the paper for publication but does not fully meet PLOS Global Public Health’s publication criteria in its current state. The manuscript requires major revision, therefore, we invite you to submit a revised version of the manuscript that addresses the points put forward by the reviewers.

The reviewers’ comments are included at the bottom of this letter, along with those of the editor who coordinated the review of your paper. Please, note that one of the reviewers had their comments in the PDF hard copy.

The reviewer(s) would like to see major revisions made to your manuscript before publication. I encourage you to attend carefully to their feedback. I believe that their suggestions, if followed, will result in a better paper. 

We look forward to receiving your revised manuscript.

Kind regards,

Joseph Adu PhD (ABD)

Academic Editor

Reviewer 1

Please, see the attached PDF hard copy for the reviewer's comments.

Reviewer 2

MANUSCRIPT ID: PONE-D-22-34478

TITLE: “Are all referrals necessary ?” Experiences and perceptions of maternity healthcare

providers on emergency intrapartum referrals in Dar es Salaam, Tanzania.

1. Recommendation (Answer options: Accept, Minor Revision, Major Revision, Reject)

MAJOR REVISION

2. Is the manuscript technically sound, and do the data support the conclusions? (Answer options: Yes, No, Partly)

3. Has the statistical analysis been performed appropriately and rigorously? (Answer options: Yes, No, I don’t know, N/A)

4. Does the manuscript adhere to the PLOS Data Policy? Additional details can be found at http://journals.plos.org/plosone/s/materials-and-software-sharing. (Answer options: Yes, No)

5. Is the manuscript presented in an intelligible fashion and written in standard English? (Answer options: Yes, No) BUT SOME PROOF READING IS REQUIRED

6. Review Comments to the Author (minimum 200 characters)

OVERALL, THE MANUSCRIPT REQUIRES MAJOR REVISION.

ABSTRACT

1. Not clear how you ensured the maximum variation sample e.g. What were some of the indicators of variation?

2. Why semi structured interview guide and not in-depth interview guide since your aim was to explore the experiences and perceptions of maternity healthcare providers on emergency intrapartum referrals.

3. Compare data collection tool ‘semi-structured interview guide’ with what was stated in the materials and methods section ‘in-depth interviews.

4. The words: experiences and perceptions could be added to the key words.

INTRODUCTION

The study provided adequate introduction to the study.

CONCEPTUAL FRAMEWORKS

Under fig 2. The statement “the illustration above shows constructs that may influence perception and experience of maternity care providers on emergency intrapartum referrals “this connotes that a figure could be referred to above, if statements are not following the fig. Directly, then rephrase the statement to exclude the word ‘above’.

MATERIALS AND METHODS

Braun and Clarke thematic analysis involves six steps as you mentioned. If this approach was used, let it be reflected in your analysis section the main steps Braun& Clarke describes in their thematic analysis- check the use of code book with Braun and Clarke, especially if a code book was used, then check your analysis steps as opposed to Braun and Clarke’s approach.

RESULTS

1. Provide a brief introduction of the major themes and how the subthemes are interwoven before you begin the details about the subthemes.

2. ‘Mcp in receiving facilities perceive mcp in referring facility as incompetent and

Irresponsible’- this was a subtheme under the main theme “limited communication between referring and receiving facilities communication” but I think it could have been a major theme on its own.

3. Results section was characterised with grammatical issues, it was not clear sometimes what the researcher was trying to communicate.

4. Most statements were winding and unclear.

5. What did you mean by ‘some had negative perceptions towards performing their responsibilities’.

“participants reported that, feedback from the receiving hospital was not always given.

Sometimes it was given in a form of report which tends tend up to the level of administration and

Not reaching to the maternal care providers” ?? please check the highlighted text for repeated word.

 

DISCUSSION

1. Below is a typical example of a winding statement:

2. ‘a study in Ghana, Ethiopia and Indonesia [11, 16, 23], also showed similar findings on Unavailability of sufficient infrastructure and number of staffs where referrals contributed by Deficiencies in working condition and environment of maternal care providers contributed a lot of stress, frustrations, fatigue and poor job satisfactions and loss of morally.’ ???? Reframe the above sentence.

3. These winding statements made it difficult to follow the discussions. A thorough proof- reading of the entire manuscript is required.

4. ‘lack of proper referral guidelines had made no uniformity in delivering of services especially giving feedbacks which is a barrier to a functional referral system [15].’

5. What is the meaning of the above statement? is there a standardised national protocol on referral systems, or there is none?

TRUSWORTHINESS

‘The study achieved rich data by involving majority of maternity HCPs involved in provision of

maternal care and referrals in the facilities that enabled giving a clear picture of what is

happening in referral system between referring facilities.’

I do not agree with this statement, selecting 11 participants from -84 refering facility and 197 staff of referral centre can not be described as majority.

Pg 18 line …7 check whether YOU MEANT ‘Thick’ or ‘Think’ descriptions.

Comparing with the main aim of the study, I do not think the statement below should be a limitation, as your study sought to explore the perceptions and experiences of maternity care providers.

‘despite all these strengths, the study had some limitations. Exploring only perceptions and experiences of maternal care providers narrowed the extend of data to be gathered as there are many other angles related to emergency intrapartum referrals which could have been left out related to mothers and their relatives’ views on the referrals ‘

I rather think the limitation could be the fact that you could have assessed or evaluated the referral guidelines to confirm your assertion of inadequate guidelines. To complete your story of intrapartum referrals. This could be explored further.

CONCLUSION

The conclusion could have involved the development or review of a referral manual or protocol, so that each party would be aware of their roles and expectations.

Journal Requirements: 

Reviewers' comments:

**Comments to the Author**

1. Is the manuscript technically sound, and do the data support the conclusions?

Reviewer #1: Yes

2. Has the statistical analysis been performed appropriately and rigorously? 

Reviewer #1: N/A

3. Have the authors made all data underlying the findings in their manuscript fully available?

Reviewer #1: No

4. Is the manuscript presented in an intelligible fashion and written in standard English?

Reviewer #1: Yes

5. Review Comments to the Author

Reviewer #1: The manuscript is well-written and covers an interesting topic.

Having read and reviewed this manuscript, I am confident it can be accepted for publication after addressing some minor changes that have been suggested across the manuscript, including reviewing some quotation styles where quotations that are less than 40 words could be in quotation marks within the text and present all quotation of 40 or more words in a block quotation.

In addition, it is suggested that the section on methodological considerations should come before the section on results and findings.

6. PLOS authors have the option to publish the peer review history of their article (what does this mean?). If published, this will include your full peer review and any attached files.

Reviewer #1: **Yes: **Jean Pierre Ndayisenga

---

## [Author Response · Author response to Decision Letter 0]

11 May 2023

Reviewer’s comment Author’s response Page and lines the changes inserted

Abstract 

Reviewer#1

The abstract is well developed and easy to understand

 Thank you for the compliment N/A

Reviewer 2

1. Not clear how you ensured the maximum variation sample eg. What were some of 

The indicators of variation?

2. Why semi structured interview guide and not in-depth interview guide since your 

aim was to explore the experiences and perceptions of maternity healthcare 

Providers on emergency intrapartum referrals.

3. Compare data collection tool ‘semi-structured interview guide’ with what was 

Stated in the materials and methods section ‘in-depth interviews.

4. The words: experiences and perceptions could be added to the key words. The purposive sampling technique ensured maximum variation of participants by recruiting participants based on the expected reasonable coverage of the participants given the purpose of the study and interest. The indicators of variation included the differences in years of experience, cadre, levels, and study sites. 

In-depth interview guide was used to explore perceptions and experiences of maternity healthcare providers on emergency intrapartum referrals. The changes have been made accordingly.

Thank you for your observation.

The changes have been made, now it reads as in-depth interview guide.

Thank you for the suggestion. 

The two words have been added as suggested. Page 7

Line 151-153

Page 8

Line 165

Page 2

Line 49-50

Introduction

Reviewer 2

The study provided adequate introduction to the study Thank you for the compliment. N/A

Conceptual framework

Reviewer#1

This section is well discussed Thank you N/A

Reviewer 2

Under fig 2. The statement “the illustration above shows constructs that may influence 

perception and experience of maternity care providers on emergency intrapartum referrals “ this 

connotes that a figure could be referred to above, if statements are not following the fig. 

Directly, then rephrase the statement to exclude the word ‘above’ Thank you for your observation. 

The word “above” has been removed. Page 4

Line 93

Materials and methods

Study design and study setting

Reviewer#1

The study settings have similarities in what? ,to complete the statement Thank you for your observation. 

The statement has been completed. Page 5

Line 122-124

Study participants and recruitment

Reviewer#1

Indicate number of each category of participants The number of participants in each category have been indicated as suggested. Page 7

Line 159-160

Data collection

Reviewer#1

The segment is well written Thank you. N/A

Reviewer 2

Braun and Clarke thematic analysis involves six steps as you mentioned. If this approach 

was used, let it be reflected in your analysis section the main steps Braun& Clarke 

describes in their thematic analysis- check the use of code book with Braun and Clarke 

esp. If a code book was used, then check your analysis steps as opposed to Braun and 

Clarke’s approach.

 Thank you for your comment. 

Authors acknowledge to have developed and used a codebook. The analysis section has been revised accordingly to incorporate the missing steps based on the Braun and Clarke’s analysis approach. Page 8,9

Line 182-194

Trustworthiness

Reviewer#2

‘The study achieved rich data by involving majority of maternity HCPs involved in

provision of

maternal care and referrals in the facilities that enabled giving a clear picture of what is

happening in referral system between referring facilities.’

I do not agree with this statement, selecting 11 participants from -84 referring facility and

197 staff of referral centre cannot be described as majority.

Pg 18 line ...7 check whether YOU MEANT ‘Thick’ or ‘Think’ descriptions.

Comparing with the main aim of the study, I do not think the statement below should be a

limitation, as your study sought to explore the perceptions and experiences of maternity

care providers.

‘despite all these strengths, the study had some limitations. Exploring only perceptions and

experiences of maternal care providers narrowed the extend of data to be gathered as there

are many other angles related to emergency intrapartum referrals which could have been

left out related to mothers and their relatives’ views on the referrals ‘

I rather think the limitation could be the fact that you could have assessed or evaluated the

referral guidelines to confirm your assertion of inadequate guidelines. To complete your

story of intrapartum referrals. This could be explored further. Thank you.

The statement has been re-phrased and the word ‘majority’ has been omitted.

Thank you. 

Word has been changed to thick.

Thank you. 

The limitation has been changed. Now the limitation statement reads as Assessment of referral guidelines was not done to confirm the assertion on adequacy of the guidelines” Page 10

Line 220-221

Page 9

Line 216

Page 10

Line 224-226

Results 

Reviewer#1

Double check and write only all participants' quotes of 40 words and above in a block and maintain quotes less than 40 words in-text Thank you.

All the quotes have been double checked and revised accordingly. Page 11-16

Reviewer#2

1. Provide a brief introduction of the major themes and how the subthemes are

interwoven before you begin the details about the subthemes.

2. ‘Mcp in receiving facilities perceive mcp in referring facility as incompetent and

Irresponsible’- this was a subtheme under the main theme “limited communication

between referring and receiving facilities communication” but I think it could have been a

major theme on its own.

3. Results section was characterised with grammatical issues, it was not clear

sometimes what the researcher was trying to communicate.

4. Most statements were winding and unclear.

5. What did you mean by ‘some had negative perceptions towards performing their

responsibilities’.

“participants reported that, feedback from the receiving hospital was not always given.

Sometimes it was given in a form of report which tends tend up to the level of

administration and

Not reaching to the maternal care providers” ?? please check the highlighted text for

repeated word. Brief introductions have been made to all the major themes

Thank you for your suggestion. Authors have revised the results section and added one theme as suggested. Now, the number of themes has changed from 2 to 3 and all necessary changes have been made throughout the manuscript. 

Grammatical issues on results section have been corrected. 

Repeated word has been deleted.

 Page 11-16

Page 11-16

Page 11-16

Discussion 

Reviewer#2 

Below is a typical example of a winding statement:

2. ‘a study in Ghana, Ethiopia and Indonesia [11, 16, 23], also showed similar findings

on Unavailability of sufficient infrastructure and number of staffs where referrals

contributed by Deficiencies in working condition and environment of maternal care

providers contributed a lot of stress, frustrations, fatigue and poor job satisfactions

and loss of morally.’ ???? Reframe the above sentence.

3. These winding statements made it difficult to follow the discussions. A thorough

proof- reading of the entire manuscript is required.

4. ‘lack of proper referral guidelines had made no uniformity in delivering of services

Especially giving feedbacks which are a barrier to a functional referral system [15].’

5. What is the meaning of the above statement? is there a standardised national

protocol on referral systems, or there is none?

 Thank you for noting this. 

The sentence has been reframed accordingly.

Proof-reading of the entire manuscript has been done as suggested. 

Thank you.

There is no standardized national referral systems as well as the guideline for referrals. Page 17

Line 402-404

Page 16-18

Full lengthy manuscript. 

Conclusion

Reviewer 1

The manuscript is well-written and covers an interesting topic. 

Having read and reviewed this manuscript, I am confident it can be accepted for publication after addressing some minor changes that have been suggested across the manuscript, including reviewing some quotation styles where quotations that are less than 40 words could be in quotation marks within the text and present all quotation of 40 or more words in a block quotation.

In addition, it is suggested that the section on methodological considerations should come before the section on results and findings. 

Thank you for the compliment.

Methodological consideration segment has been moved to the methodology section just before the results section. Page 9

Line 206-226

Reviewer#2

The conclusion could have included development or review of a referral manual or

protocol, so that each party would be aware of their roles and expectations Thank you for the suggestion. 

The point has been added,

‘Standard referral guideline should be developed, so that each hospital would be aware of their roles and expectations’ Page 19

Line 443-445

---

## [Editor Report · Decision Letter 1]

15 May 2023

PONE-D-22-34478R1“Are all referrals necessary ?” Experiences and perceptions of maternity healthcare providers on emergency intrapartum referrals in Dar es Salaam, Tanzania.PLOS ONE

Dear Dr. Selemani,

Thank you for submitting your manuscript to PLOS ONE. After careful consideration, we feel that it has merit but does not fully meet PLOS ONE’s publication criteria as it currently stands. Therefore, we invite you to submit a revised version of the manuscript that addresses the points raised during the review process.

ACADEMIC EDITOR: 

Thank you for taking time to respond to the reviewers comments. However, the article could further benefit from another proofreading reading to correct minor errors and typos. For instance, on line 412 of the unmarked copy: You stated; In our study some of providers....... The 'of' is not needed. I know it is a typo! There are many others that run through your work. Kindly, take your time to correct these minor errors and typos. Also, check line 32 of the abstract and reframe the statement on in-depth interview guide.

Again, please review your reference list to ensure that it is complete and correct. If you have cited papers that have been retracted, remove these references and replace them with relevant current references.

We look forward to receiving your revised manuscript.

Kind regards,

Joseph Adu, MSc., Mphil

Academic Editor

PLOS ONE
---

## [Author Response · Author response to Decision Letter 1]

12 Jul 2023

Thank you for the comment received.

Figure 2 has been referred in the text.

I will be grateful to hear from you.

---

## [Editor Report · Decision Letter 2]

14 Jul 2023

PONE-D-22-34478R2“Are all referrals necessary ?” Experiences and perceptions of maternity healthcare providers on emergency intrapartum referrals in Dar es Salaam, Tanzania.PLOS ONE

Dear Dr. Shekha,

Thank you for submitting your manuscript to PLOS ONE. After careful consideration, we feel that it has merit but does not fully meet PLOS ONE’s publication criteria as it currently stands. Therefore, we invite you to submit a revised version of the manuscript that addresses the points raised during the review process.

We look forward to receiving your revised manuscript.

Kind regards,

Joseph Adu, PhD, MSc., Mphil

Academic Editor

PLOS ONE

Journal Requirements:

Additional Editor Comments:

Having scanned through your manuscript, it seems you did not adhere to my earlier suggestion that the paper needs thorough proofreading and editing. This is because the paper is still full grammatical errors. Kindly, present the proofreading in track changes.Please, note that until the quality of the paper improves PLOS ONE cannot publish it. Also, check all references per the journal guidelines to avoid any further delay.
---

## [Author Response · Author response to Decision Letter 2]

20 Sep 2023

proofreading of the entire work has been done

reviewers comments on grammatical errors have been addressed

References have been revised

Fig 1 has been referred as suggested

---

## [Editor Report · Decision Letter 3]

27 Sep 2023

PONE-D-22-34478R3“Are all referrals necessary ?” Experiences and perceptions of maternity healthcare providers on emergency intrapartum referrals in Dar es Salaam, Tanzania.PLOS ONE

Dear Dr. Shekha Selemani,

Thank you for submitting your manuscript to PLOS ONE. After careful consideration, we feel that it has merit but does not fully meet PLOS ONE’s publication criteria as it currently stands. Therefore, we invite you to submit a revised version of the manuscript that addresses the points raised during the review process.

ACADEMIC EDITOR: Comments

The authors of this manuscript have been given at least two chances to proofread the paper for grammatical errors and typos, but a critical look at the final version shows the paper cannot be accepted for publication. That is, there are cases where the language is unclear and a bit difficult to follow. I advise the authors work with a copyeditor to improve the flow of the text for easy readability. The revised edition should be submitted within two weeks for consideration. Failure to proofread the manuscript thoroughly will result in the rejection of the paper. I have included some of the errors identified:

Lines 32-34: We used an in-depth interview guide to conduct eleven depth interviews. The thematic analysis approach by Braun and Clarke (2006) guided the data analysis. There is something wrong with this statement.

Lines 77-78: The woman been as described as “not my case” [9]. Look at this statement again.

Line 82: A qualitative study ,by Afari in Ghana showed that maternity care providers perceived some challenges related….. Is the comma between study and by necessary?

Line 212: Four criteria as explained by Shenton AK, et al. were used to assess the….

I believe this should read Shenton et al. in addition to the superscript.

SOPs runs through the manuscript but the authors failed to define the acronym.

We look forward to receiving your revised manuscript.

Kind regards,

Joseph Adu, PhD.

Academic Editor

PLOS ONE
---

## [Author Response · Author response to Decision Letter 3]

7 Jan 2024

Line 32-34, 77-78, 82 and 212 have been revised

SOPs has been defined as Standard Operating Procedures

References have been revised to ensure they are complete and correct. 3 more references have been added.

Proofreading has been done with native English speaker.

---

## [Editor Report · Decision Letter 4]

20 Jan 2024

“Are all referrals necessary ?” Experiences and perceptions of maternity healthcare providers on emergency intrapartum referrals in Dar es Salaam, Tanzania.

PONE-D-22-34478R4

Dear Dr. SELEMANI,

We’re pleased to inform you that your manuscript has been judged scientifically suitable for publication and will be formally accepted for publication once it meets all outstanding technical requirements.

Kind regards,

Hannah Tappis, DrPH, MPH

Academic Editor

PLOS ONE
---

## [Editor Report · Acceptance letter]

12 Feb 2024

PONE-D-22-34478R4 

PLOS ONE

Dear Dr. Selemani, 

I'm pleased to inform you that your manuscript has been deemed suitable for publication in PLOS ONE. Congratulations! Your manuscript is now being handed over to our production team.

Kind regards, 

on behalf of

Dr. Hannah Tappis 

Academic Editor

PLOS ONE